# LinMU: Multimodal Understanding Made Linear

**Hongjie Wang**                                                                *hongjiewang@princeton.edu*
*Department of Electrical and Computer Engineering*
*Princeton University*

**Niraj K. Jha**                                                                        *jha@princeton.edu*
*Department of Electrical and Computer Engineering*
*Princeton University*

**Reviewed on OpenReview:** *https://openreview.net/forum?id=6BYdTSNrab*

## Abstract

Modern Vision-Language Models (VLMs) achieve impressive performance but are limited by the quadratic complexity of self-attention, which prevents their deployment on edge devices and makes their understanding of high-resolution images and long-context videos prohibitively expensive. To address this challenge, we introduce LinMU (Linear-complexity Multimodal Understanding), a VLM design that achieves linear complexity for the language model decoder without using any quadratic-complexity modules while maintaining the performance of global-attention-based VLMs. LinMU replaces every self-attention layer in the language model decoder with an M-MATE block: a dual-branch module that combines a bidirectional state-space model for global context (Flex-MA branch) with localized Swin-style window attention (Local-Swin branch) for adjacent correlations. To transform a pre-trained VLM into the LinMU architecture, we propose a three-stage distillation framework that (i) initializes both branches with self-attention weights and trains the Flex-MA branch alone, (ii) unfreezes the Local-Swin branch and fine-tunes it jointly with the Flex-MA branch, and (iii) unfreezes the remaining blocks and fine-tunes them using LoRA adapters, while regressing on hidden states and token-level logits of the frozen VLM teacher. On MMMU, TextVQA, LongVideoBench, Video-MME, and other benchmarks, LinMU matches the performance of teacher models, yet reduces Time-To-First-Token (TTFT) by up to $2.7\times$ and improves token throughput by up to $9.0\times$ on minute-length videos. Ablations confirm the importance of each distillation stage and the necessity of the two branches of the M-MATE block. We also conduct distillation on various VLM backbones to validate the universality of LinMU. The proposed framework demonstrates that state-of-the-art multimodal reasoning can be achieved without quadratic attention, thus opening up avenues for long-context VLMs that can deal with high-resolution images and long videos.

## 1 Introduction

Recent advances in Vision-Language Models (VLMs) (Liu et al., 2023; Li et al., 2024; Chen et al., 2024d; Wang et al., 2024b; Liu et al., 2025) have delivered outstanding multimodal understanding performance across a wide range of image- and video-centric benchmarks, often matching or surpassing fully-supervised systems through only few-shot prompting. VLMs have been widely deployed in various domains, including but not limited to medicine (Tanno et al., 2025), remote sensing (Kuckreja et al., 2024), robotics (Kim et al., 2024), autonomous driving (Jiang et al., 2025), agriculture (Arshad et al., 2025), and science education (Lu et al., 2022). However, these achievements are accompanied by substantial computational overhead. Since global self-attention scales quadratically with the number of tokens, processing minute-length, high-resolution videos requires aggressive frame sampling or clusters of high-end GPUs, thus limiting the practical deployment of current state-of-the-art models and motivating research into linear-complexity alternatives.

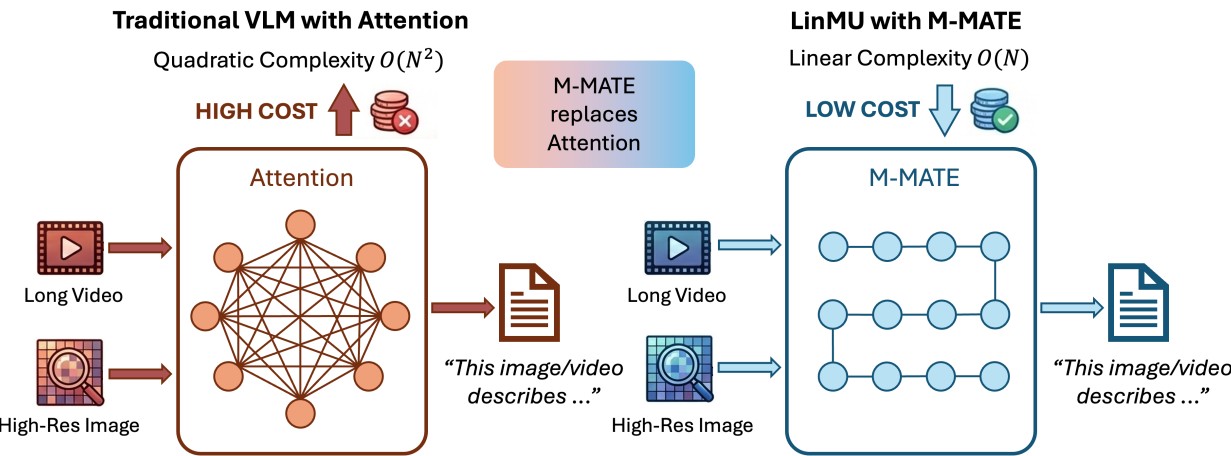

Figure 1: By replacing quadratic-complexity attention layers with our proposed M-MATE layers, LinMU achieves linear complexity without sacrificing model performance.

In this article, we seek to break the quadratic barrier for multimodal understanding. We propose **LinMU (Linear-complexity Multimodal Understanding)**, a framework that replaces every self-attention layer in a VLM with a linear-complexity module while preserving model performance, as shown in Fig. 1. Inspired by the recently introduced MATE block in LinGen (Wang et al., 2025b;a), we propose the M-MATE (Multimodal-MATE) block, a linear-cost replacement for self-attention, as the building block of a multimodal Transformer. The M-MATE block consists of two complementary branches: (i) a Flex-MA branch based on the masked bidirectional Mamba2 model (Dao & Gu, 2024) for capturing long-range dependencies efficiently, and (ii) a Local-Swin branch using 3D Swin attention (Liu et al., 2022) with a fixed small window size to focus on short-range spatiotemporal correlations. Each M-MATE block runs in $O(N)$ time with respect to token count, inheriting linear scalability from the state-space model (SSM) and fixed-size window operations. By replacing self-attention with M-MATE throughout the network, LinMU can handle significantly longer contexts than traditional Transformers for a given compute budget.

Adopting a fully linear architecture risks degrading accuracy. Hence, we employ a careful knowledge distillation strategy to transfer the capabilities of a pretrained attention-based teacher model to LinMU. Specifically, we distill from a strong VLM teacher to a LinMU student by reusing the teacher's attention weights to initialize our M-MATE branches and then fine-tuning the student model in stages. We design a three-stage distillation pipeline: (1) train the Flex-MA branch alone to mimic the teacher's global attention behavior, (2) train the Local-Swin branch together with the Flex-MA branch to recover local correlation modeling, and (3) jointly fine-tune the entire M-MATE-augmented model (with other backbone layers tuned via low-rank adapters) to close any remaining performance gap. During distillation, we optimize a combination of token-level and sequence-level losses, aligning the student's predictions with the teacher's outputs (in the spirit of cross-entropy with teacher soft targets and using teacher-generated sequences as pseudo-labels) (Wang et al., 2024a). We also include a hidden feature alignment loss in the distillation objective to enable faster convergence. This staged distillation framework ensures a smooth transition from quadratic-complexity attention to linear-complexity M-MATE blocks without sacrificing understanding capability. We summarize our contributions as follows:

- We present LinMU, the first multimodal (vision-language) model that achieves linear complexity in input length by replacing self-attention with the M-MATE block without sacrificing model performance. LinMU enables long-context video and image understanding (potentially minutes of video) on modest hardware, providing much better scalability than the standard VLMs.

- We develop a multi-stage distillation framework to transfer knowledge from a standard Transformer-based VLM into our linear architecture. By reusing the teacher's projection weights to initialize

a masked bidirectional Mamba2 model (Flex-MA branch) and a 3D Swin-attention (Local-Swin branch), and by employing carefully designed distillation losses, we preserve the accuracy of the original teacher model to a great extent.

- We demonstrate, using challenging image and video understanding benchmarks, that LinMU achieves performance on par with the teacher models (i.e., NVILA-8B-Video and Qwen2.5-VL-7B-Instruct) while significantly improving efficiency. In our experiments, LinMU maintains competitive performance on tasks like MMMU, LongVideoBench, and Video-MME, and enables markedly faster inference in terms of TTFT and token throughput, thus validating linear scalability. We also present ablation studies that quantify the impact of each distillation stage and loss term, and confirm the importance of M-MATE components (Flex-MA and Local-Swin branches) in maintaining teacher performance.

Overall, LinMU illustrates that we can make multimodal Transformers linear without compromising their remarkable understanding capabilities. This opens the door to scaling up context lengths (e.g., using more video frames or higher-resolution features) and deploying VLMs in real-time or resource-constrained settings, which was previously infeasible due to quadratic costs.

We organize the remainder of this article as follows. Section 2 surveys recent progress on VLMs and prior efforts to make them more efficient. Section 3 introduces the LinMU framework, detailing the M-MATE block and our three-stage distillation pipeline. Section 4 presents comprehensive experiments to evaluate the performance and efficiency of LinMU, along with ablation studies that target the distillation stages, loss terms, and block components. Finally, Section 6 concludes the article and outlines directions for future work.

## 2 Related Work

In this section, we situate our contribution within three complementary lines of research. First, we review the evolution of modern VLMs, with emphasis on the high-capacity Transformer-based architectures (Section 2.1). Second, we discuss methods that improve the efficiency of VLMs primarily through token pruning and structural compression while keeping the underlying attention backbone unchanged (Section 2.2). Third, we survey linear-complexity alternatives to attention, including state-space and recurrent architectures and their deployment on VLMs, which are most closely related to our proposed framework (Section 2.3).

### 2.1 Large Vision-Language Models

Recent years have seen rapid progress in large VLMs, which couple a high-capacity language backbone with a visual encoder through a multimodal projector and stacks of Transformer attention layers. Early models, such as Flamingo (Alayrac et al., 2022), BLIP-2 (Li et al., 2023a), MiniGPT-4 (Zhu et al., 2023), and LLaVA (Liu et al., 2023), established the now-standard recipe of using a frozen or lightly-tuned vision encoder to produce visual tokens that are injected into a frozen or instruction-tuned large language model (LLM).

Contemporary open-source models push this paradigm to higher capacity and broader modalities. Qwen2-VL (Wang et al., 2024b) and the more recent Qwen2.5-VL (Bai et al., 2025) series extend the Qwen language backbone with a streamlined encoder in the Visual Transformer (ViT) style and improved visual projector, supporting images, documents, and videos at native resolution and long contexts. Qwen2.5-VL achieves competitive performance on diverse benchmarks, such as MMMU (Yue et al., 2024) and Math-Vista (Lu et al., 2023), while remaining fully Transformer-based, with quadratic attention in the LLM decoder.

The LLaVA family follows a similar design but is trained almost entirely from open data. LLaVA-NeXT (Liu et al., 2024b) improves reasoning, optical character recognition (OCR), and world knowledge over LLaVA-1.5 (Liu et al., 2024a) and extends to video understanding. LLaVA-OneVision (Li et al., 2024) further pushes performance by training on native-resolution images and providing a complete open training pipeline.

Parallel to these, NVILA (Liu et al., 2025) introduces a "scale-then-compress" paradigm designed to optimize the efficiency-accuracy frontier. Unlike traditional models that trade off resolution for speed, NVILA first scales up spatial and temporal inputs to capture fine-grained details and then compresses these visual tokens to maintain high throughput during inference. Meta's Llama-3.2-Vision models (Dubey et al., 2024) are optimized for high-accuracy image reasoning and document understanding with long context windows. InternVL-2.5 (Chen et al., 2024c) optimizes the visual encoder (InternViT) specifically for semantic alignment, which can reduce the number of visual tokens needed to convey the same amount of information compared to vanilla CLIP encoders.

As discussed above, recent VLMs tend to explore unified architectures that handle images, documents, and videos through scaling of parameter-count and context length. However, across almost all of these models, the multimodal backbone is still built from standard self- and cross-attention Transformers. Hence, both computation and memory scale quadratically with the total number of tokens, which becomes a growing bottleneck as models move to high-resolution, multi-image, and video inputs.

### 2.2 Efficient Vision-Language Models with Attention

To mitigate the cost of quadratic attention without changing the core architecture, a large body of work focuses on improving the efficiency of VLMs through token reduction, structural pruning, and other model-compression techniques. These methods can be categorized into strategies that operate on the token sequence (e.g., pruning and compression), network structure (e.g., layer pruning, distillation), and at system level (e.g., quantization, KV-cache optimizations).

**Visual token pruning and compression.** A dominant line of work observes that visual tokens are heavily redundant relative to text tokens and seeks to reduce the number of vision tokens that reach the LLM. FastV (Chen et al., 2024b) proposes a plug-and-play inference procedure that discards roughly half of the image tokens after the second layer, yielding substantial speedups while incurring minimal accuracy loss. Instruction-Guided Visual Token Pruning (Huang et al., 2024) introduces a group-wise token pruning module based on attention rollout, using the instruction to guide which vision tokens are pruned both within the vision encoder and the LLM. LLaVA-PruMerge (Shang et al., 2025) proposes an adaptive reduction of visual tokens using the sparsity of class-token attention. LLaVolta (Chen et al., 2024a) introduces a Visual Context Compressor and a new training scheme, demonstrating that up to 70% of visual tokens can be removed during testing while only incurring minor degradation. SparseVLM (Zhang et al., 2024a) proposes a text-guided, training-free pruning mechanism that ranks visual tokens using self-attention with selected text tokens and adaptively determines sparsification ratios per layer. Related works, such as FEATHER (Endo et al., 2025) and CoViPAL (Tang et al., 2025), refine these ideas by considering layer-wise token importance and contextual dependencies. PruneVid (Huang et al., 2025) and LLaVA-Scissor (Sun et al., 2025) further deploy the token pruning idea to Video-centric VLMs.

**Architectural compression.** Beyond token-level sparsification, several works compress VLMs at the architecture level. Short-LVLM (Ma et al., 2025) proposes a new pruning framework that identifies redundant layers while compensating for cross-modal feature changes, providing training-free compression that is model-agnostic and compatible with token-level methods. ECoFLaP (Sung et al., 2023) proposes a coarse-to-fine layer-wise pruning scheme for large VLMs that first computes a global importance score via zeroth-order gradient approximations to assign adaptive sparsity ratios across layers, and then performs local unstructured pruning within each layer. Complementary to pruning, various works distill the visual representation space of large VLM teachers (e.g., CLIP ViT-L/14) into lightweight students using small- or mid-scale datasets, with a particular focus on preserving relative geometry in the joint vision–language embedding space to maintain open-vocabulary out-of-distribution generalization (Li et al., 2023b).

**Quantization and key-value (KV) cache optimization.** Another line of efficiency work targets numerical precision and cache storage rather than architecture. NVILA employs quantization in both the vision encoder and the LLM backbone to reduce TTFT and improve token throughput. MBQ (Modality-Balanced Quantization) (Li et al., 2025) observes that vision and language tokens in large VLMs have very different sensitivity to quantization noise and thus uses gradient-based sensitivity indicators for each modality. ReKV (Di et al., 2025) proposes a training-free approach that enables efficient streaming video question-

answering via in-context video window KV-cache retrieval. However, such a pure window-based perception pattern for inputs incurs performance degradation.

In summary, most of the current efficiency efforts for VLMs retain the Transformer attention backbone. While they dramatically reduce constant factors, the underlying complexity with respect to the remaining tokens remains quadratic, and the attention mechanism itself is unchanged. Our work is complementary: Instead of compressing the token sequence or pruning layers on top of attention, we investigate a linear-complexity module that can directly replace attention layers in existing VLM backbones.

### 2.3 Linear-Complexity Alternatives to Attention in VLMs

To address the quadratic complexity of attention, one line of work explores sparse or factorized attention to handle longer inputs. In vision, local window attention and factorized space-time attention have been used in video Transformers, such as Vision Longformer (Zhang et al., 2021) and Video Swin Transformer (Liu et al., 2022), to reduce the cost of global attention. These approaches limit each token's receptive field (e.g., attending only within a local patch or frame window), which yields linear or near-linear complexity per layer but often requires adding some global layers to not hurt performance. For example, Video Swin Transformer achieves linear scaling by restricting attention to non-overlapping windows, then shifting windows between layers to propagate information. Such designs can be very effective, but purely local attention cannot capture long-range dependencies unless some global mechanism is present. For natural language processing, BigBird (Zaheer et al., 2020) and Longformer (Beltagy et al., 2020) similarly use sparse attention patterns (sliding windows, random or global tokens) to reach approximate linear complexity, with trade-offs in completeness of context modeling. In contrast, LinMU's M-MATE block provides a global sequence mixing through the MA-Flex branch and local neighbor focus through the Local-Swin branch, together achieving full coverage of dependencies with linear cost.

Another line of work explores different computational patterns to achieve linear complexity, including linear attention (Katharopoulos et al., 2020), SSMs (Gu et al., 2021; Gu & Dao, 2024), and recurrent architectures (Peng et al., 2023; Sun et al., 2023). They have been explored for language modeling and other modalities, but are hardly competitive with Transformers at scale without involving quadratic-complexity layers.

Specifically, Mamba (Gu & Dao, 2024) introduces a selective SSM architecture that conditions the SSM parameters on the input and implements a hardware-aware recurrent scan, achieving linear complexity in sequence length and significantly higher throughput than Transformers for long sequences. Its successor Mamba2 (Dao & Gu, 2024) has the attention format and is compatible with the efficient attention kernels, such as FlashAttention (Dao et al., 2022) and XFormers (Lefaudeux et al., 2022). Building on this, VMamba (Liu et al., 2024c) adapts Mamba to visual data by introducing Visual State-Space blocks and a 2D Selective Scan module. VMamba traverses images along multiple scan directions to capture global context with linear complexity. Subsequent works, such as Multi-Scale VMamba (Shi et al., 2024) and Spatial-Mamba (Xiao et al., 2025), refine scanning strategies and multi-scale context aggregation.

Several recent works apply linear-time architectures directly to multimodal learning and VLMs. VL-Mamba (Qiao et al., 2024) replaces the Transformer LLM with a pretrained Mamba language model and introduces a MultiModal Connector with a Vision Selective Scan module, demonstrating competitive performance with Transformer-based multimodal LLMs while enjoying linear scaling in sequence length and faster inference. VisualRWKV (Hou et al., 2025b) extends the RWKV linear recurrent neural network to multimodal learning, proposing data-dependent recurrence, a "sandwich" visual prompting scheme, and a 2D image scanning mechanism to handle visual tokens.

To obtain better performance, hybrid architectures that interleave quadratic-complexity attention and linear-complexity SSM layers are beginning to appear in both text-only and multimodal settings. Nemotron-Nano-2 models (Basant et al., 2025) replace some of the self-attention layers of a Transformer with Mamba-2 layers, achieving higher throughput on long-context reasoning while maintaining Transformer-level accuracy. Nemotron-Nano-2 VL (Deshmukh et al., 2025) extends this hybrid Transformer-Mamba design to vision-language tasks, such as video understanding and document intelligence, demonstrating that linear-time modules can be integrated into competitive multimodal systems.

As discussed above, most existing linear-complexity VLMs typically take one of two approaches: (i) replacing attention layers of the entire backbone with a linear-complexity alternative, e.g., Mamba or RWKV, which often lags behind Transformer-based VLMs on complex vision understanding tasks due to the **adjacency preservation issue** (see Sec. 3.1); or (ii) hybrid architectures that interleave attention layers with linear-complexity layers, which still suffers from overall quadratic complexity. By contrast, LinMU replaces all attention layers in Transformer-based VLMs without compromising performance, especially on complex vision tasks. By building on the M-MATE block design and a tailored distillation schedule, we show that such a replacement is indeed possible for pre-trained VLMs on complex vision-language understanding tasks.

## 3 Methodology

In this section, we first describe our proposed architecture, LinMU, and then illustrate the distillation framework to transform a global-attention-based pre-trained VLM into LinMU.

### 3.1 The LinMU Architecture

Despite the efficiency advantages of linear sequence layers, as in Mamba and RWKV, a known challenge is applying them to large-scale vision tasks. Given the relatively better hardware efficiency of Mamba models among the linear sequence layers (Dao & Gu, 2024), we focus on Mamba in the rest of this article. When Mamba is fed image or video tokens (flattened from 2D/3D), it struggles with the **adjacency preservation issue**: Spatially and temporally adjacent tokens become non-neighboring in the flattened sequence, causing the model's performance to degrade due to the inherent correlation precision decay over long distances (Wang et al., 2025b). Prior works attempt to alleviate this problem through better token ordering (Hu et al., 2024; He et al., 2024) or by mixing some regular attention layers with SSM layers (Basant et al., 2025). However, these solutions either do not fully fix the adjacency preservation issue at a large scale or reintroduce quadratic components.

Inspired by the MATE block in LinGen (Wang et al., 2025b), we propose replacing the self-attention block in VLMs with an M-MATE block. As shown in Fig. 2, LinMU is an auto-regressive VLM composed of two parts: Vision Encoder and Token Processor (i.e., LM Decoder). Because the implementation of the Vision Encoder could be quite different across variant VLM backbones, to enable generalization, we leave the Vision Encoder unchanged and replace attention layers in the Token Processor with our proposed M-MATE blocks.

The proposed M-MATE block builds on two branches: (I) The **Flex-MA branch** that uses bidirectional Mamba2 for its linear complexity and throughput benefits. By applying a mask in the reverse direction, it flexibly achieves causal correlations and global correlations for text tokens and vision tokens, respectively. We mitigate adjacency preservation via Rotary Major-Scan (RMS) proposed in LinGen. (II) The **Local-Swin branch** that uses 3D local Swin attention to directly tackle the adjacent correlations in modeling immediate neighbor patches. The windows are fixed at a small size because it only needs to model adjacent correlations, which keeps the complexity of this branch linear and makes it very efficient. In this way, LinMU inherits the best of both worlds: global linear mixing and local precise attention.

#### 3.1.1 Flex-MA Branch

This branch is built on Mamba2 (Dao & Gu, 2024), an SSM designed as a linear-complexity alternative to attention. We choose Mamba2 for its linear-time complexity and its Transformer-format parameterization. The latter makes it compatible with modern efficient attention kernels, such as FlashAttention and xFormers. In addition, this Transformer format makes it easier to reuse attention weights in pre-trained models for initialization.

In this branch, the input token sequence (e.g., a sequence of visual and textual embeddings) is first rearranged by an RMS operation (Wang et al., 2025b). RMS permutes the tokens in a way that enables spatially or temporally neighboring vision tokens to remain closer in the sequence (mitigating the adjacency preservation issue of vanilla Mamba). The main difference between RMS and other existing special scan methods is that it achieves this with only tensor reshaping operations and is much more friendly to hardware, thus is much more efficient and incurs much less extra latency (Wang et al., 2025b). RMS only changes the order of vision

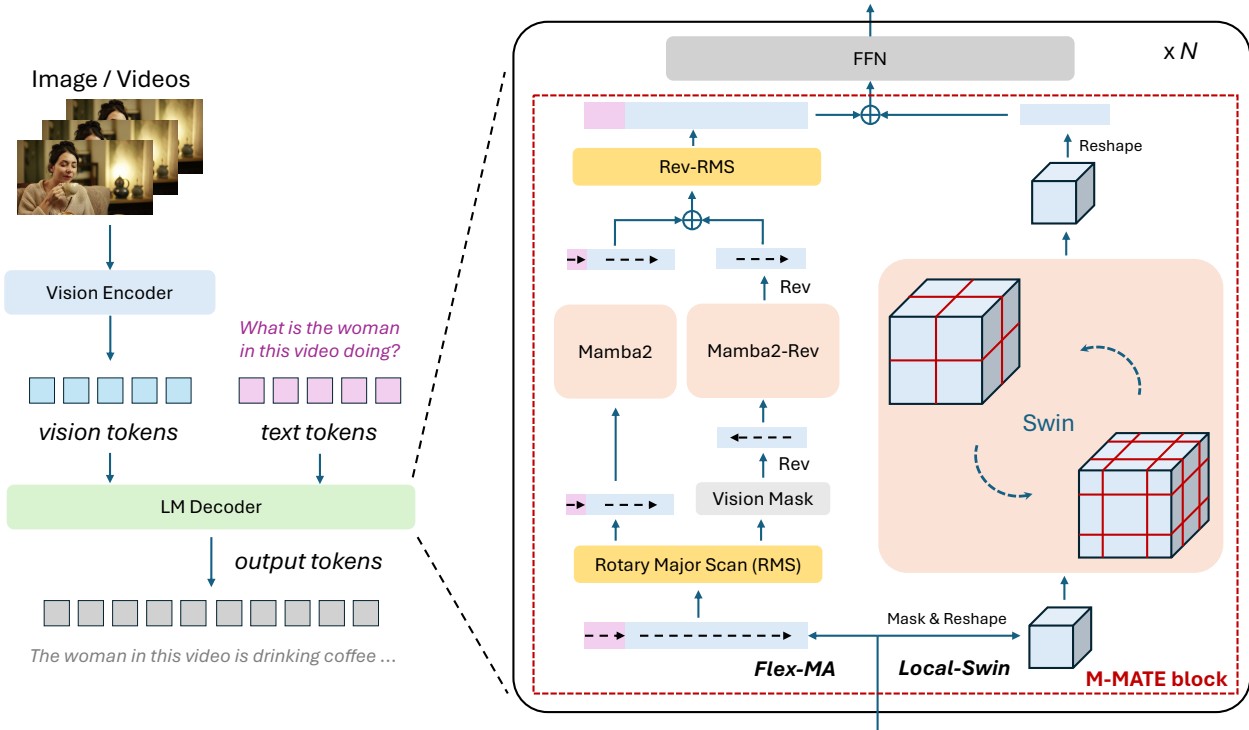

Figure 2: The overview of LinMU, in which the Vision Encoder is untouched but all the attention layers in the Language Model (LM) Decoder are replaced with our proposed M-MATE blocks. Residual/gate connections are not shown in this figure for brevity. The M-MATE block contains two parallel branches: (1) an **Flex-MA branch** built around a bidirectional Mamba2 augmented with reverse direction token mask and RMS token rearrangement, and (2) a **Local-Swin branch** that implements a local 3D Swin Attention with a fixed window size. The Flex-MA branch provides efficient long-range sequence mixing with linear complexity, while the Local-Swin branch focuses on adjacent spatial/temporal correlations to preserve local consistency. Both branch outputs are summed to produce the M-MATE block output that replaces the original self-attention output. Other parts of the LM Decoder (e.g., multi-layer perceptron, layer norms, etc.) remain unchanged.

tokens. Assume the vision token tensor shape in the latent space is $T \times H \times W$ (where $T = 1$ if it represents an image), the index of token $T[t][y][x]$ ($t$ is always 0 for an image) in the rearranged 1D sequence in the $l$-th layer is given by

$$
n_l = \begin{cases}
t \cdot (H \cdot W) + y \cdot W + x, & \text{if } l \mod 4 = 0 \\
t \cdot (H \cdot W) + x \cdot H + y, & \text{if } l \mod 4 = 1 \\
y \cdot (T \cdot W) + x \cdot T + t, & \text{if } l \mod 4 = 2 \\
x \cdot (T \cdot H) + y \cdot T + t, & \text{if } l \mod 4 = 3
\end{cases}
$$

Four different scanning patterns alternate in consecutive layers. When dealing with images, it reduces to two different patterns from four. The permuted sequence is then fed to a bidirectional Mamba2 layer, which computes an output sequence of the same length. Neither like the original (causal) Mamba that was designed for unidirectional language modeling, nor the pure bidirectional version that produces a full "attention-like" correlation map for vision tasks, we employ a vision-token mask $\mathcal{M}_V$ in the reverse direction, flexibly producing causal maps for text tokens and full maps for vision tokens, as shown in Fig. 3.

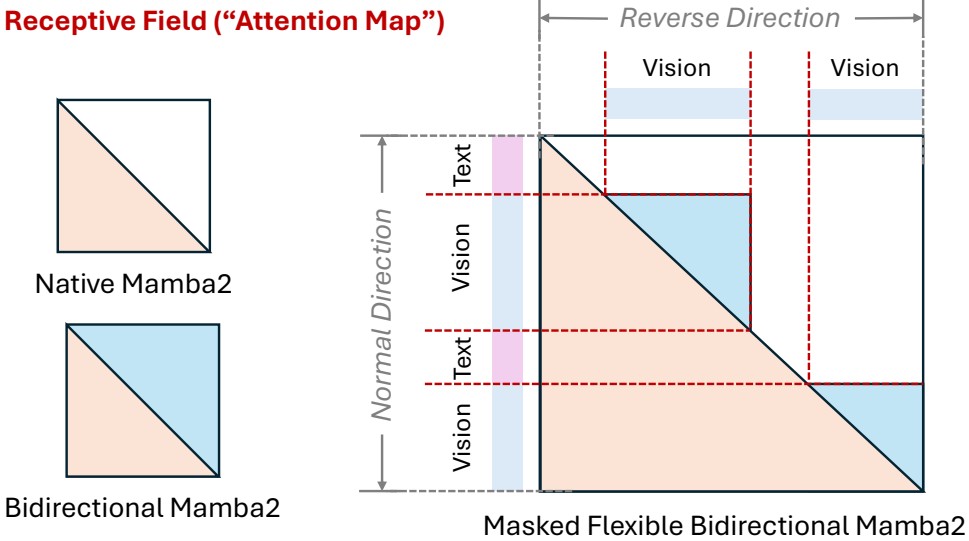

Figure 3: The receptive field comparison between native Mamba2 (suitable for text-only tasks), bidirectional Mamba2 (suitable for vision-only tasks), and our proposed masked flexible bidirectional Mamba2 (suitable for multimodal tasks).

The masked bidirectional Mamba2 performs global, linear-time sequence mixing via recurrent state updates run forward and backward. It does not compute explicit token-to-token similarity matrices but attains a full-context receptive field suitable for multimodal understanding tasks. Importantly, the computation cost of Mamba2 scales as $O(Nd^2)$ (where $N$ is the number of tokens and $d$ is the dimension of embeddings), which is linear in $N$ (Dao & Gu, 2024). The outputs of Mamba2 are finally inverse-permuted back (to undo RMS) and passed on. Overall, the Flex-MA branch is responsible for capturing global dependencies (even across hundreds of thousands of tokens) at low cost, albeit in a somewhat blurred fashion due to the limited precision of SSMs for very long ranges.

Let $\mathbf{u}_t \in \mathbb{R}^d$ be the token with index $t$ in the sequence after RMS permutation. A selective SSM layer defines input-conditioned parameters $(\mathbf{A}_t, \mathbf{B}_t, \mathbf{C}_t)$ and updates as follows:

$$\mathbf{h}_t^{\rightarrow} = \mathbf{A}_t \mathbf{h}_{t-1}^{\rightarrow} + \mathbf{B}_t \mathbf{u}_t, \quad \mathbf{y}_t^{\rightarrow} = \mathbf{C}_t \mathbf{h}_t^{\rightarrow}, \tag{1}$$

$$\mathbf{h}_t^{\leftarrow} = \tilde{\mathbf{A}}_t \mathbf{h}_{t+1}^{\leftarrow} + \tilde{\mathbf{B}}_t (\mathcal{M}_V \odot \mathbf{u}_t), \quad \mathbf{y}_t^{\leftarrow} = \tilde{\mathbf{C}}_t \mathbf{h}_t^{\leftarrow}. \tag{2}$$

Note that $\mathcal{M}_V$ is a vision token mask that retains only vision tokens. We fuse the directions with a learned gate:

$$\mathbf{y}_t^{\text{Flex}} = \mathbf{W}_o \big( \mathbf{g}_t \odot \mathbf{h}_t^{\rightarrow} + (1 - \mathbf{g}_t) \odot \mathbf{h}_t^{\leftarrow} \big). \tag{3}$$

### 3.1.2 Local-Swin Branch

In parallel with the Flex-MA branch, each M-MATE block includes a local 3D swin attention module, as shown in Fig. 4. It focuses on vision tokens and masks out text tokens. This is a form of multi-head attention restricted to local neighborhoods spatially and temporally, inspired by Swin Transformer's shifted window approach (Liu et al., 2022). In practice, given the tokens arranged in a 3D index (frame $\times$ height $\times$ width for video, or $1 \times$ height $\times$ width for images), the Local-Swin branch divides them into small windows (e.g., $16 \times 4 \times 4$ for long videos) and computes standard attention only among tokens within each window. Because window size is fixed and very small (we use $16 \times 4 \times 4$ volumes, for example), this attention is linear in $N$ (each token attends to a constant number of neighbors). The windows are shifted in consecutive layers (similar to Swin) to allow cross-window connections over multiple M-MATE blocks. The Local-Swin branch thus excels at modeling short-range and medium-range correlations: It ensures that immediately

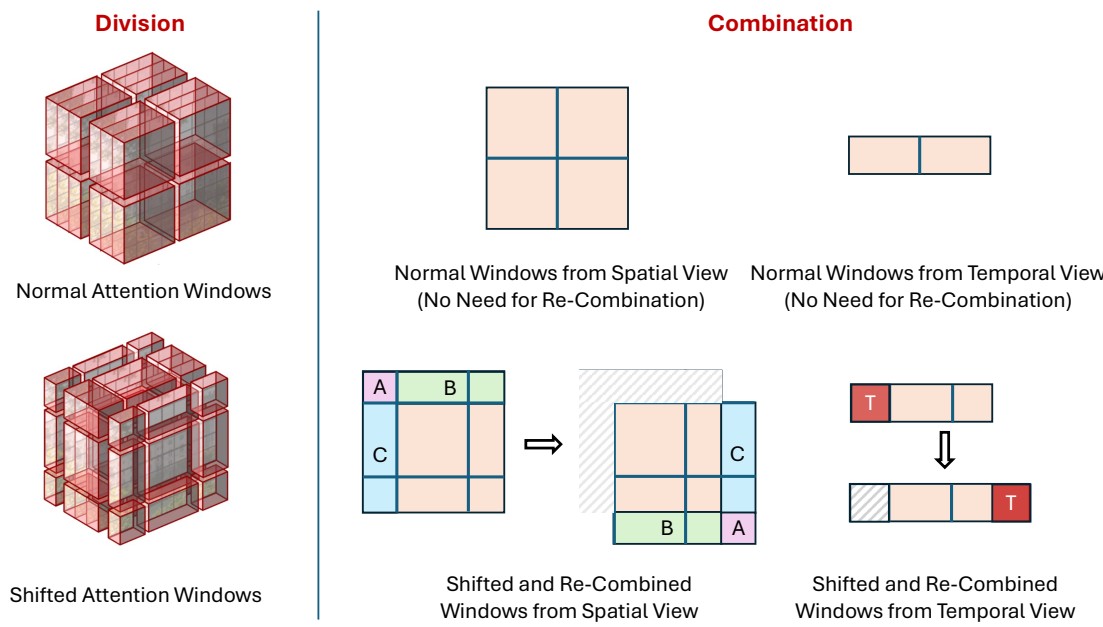

Figure 4: Window division, shifting, and re-combination in the local 3D Swin attention. After window shifting, tokens around the image/video edge are combined to form new attention windows, ensuring the total number of windows remains constant.

adjacent pixels/patches and nearby frames are consistently processed, something the Flex-MA branch may not guarantee due to token rearrangement to 1D sequences and long-range decay.

In the Local-Swin branch, vision tokens are reshaped to $\mathbb{R}^{F \times H \times W \times d}$. For window size $(\tau, s, s)$ and stride equal to window (non-overlap), we compute multi-head attention locally inside each $(\tau \times s \times s)$ block as follows:

$$\text{Attn}_{\text{win}}(\mathbf{Q}, \mathbf{K}, \mathbf{V}) = \text{softmax}\left(\frac{\mathbf{Q}\mathbf{K}^\top}{\sqrt{d_h}}\right)\mathbf{V}.$$

In alternating layers, we use a shift of $(\lfloor \tau/2 \rfloor, \lfloor s/2 \rfloor, \lfloor s/2 \rfloor)$ to enable cross-window links. The per-layer cost is $O(N \cdot \tau^2 s^4 d)$, i.e., linear in $N$ for fixed local window size.

The output of this branch is fused with the Flex-MA branch output with a learned weight to produce the final M-MATE block output, as shown in Alg. 1. By design, the M-MATE output has the same shape as a standard attention output. Hence, it can be fed into the subsequent feed-forward network (FFN) and residual connections exactly as in the original Transformer layer.

Crucially, both branches of the M-MATE block operate in parallel, and their combined cost is still linear. The cost of the masked bidirectional Mamba2 is $O(N)$ in sequence length, and the cost of the local 3D Swin attention is also $O(N)$ (with the fixed window size). By replacing all $L$ self-attention layers in a Transformer with the proposed M-MATE block, the overall complexity w.r.t. token number becomes $O(L \cdot N)$, instead of $O(L \cdot N^2)$, enabling dramatic speedups for long sequences.

## 3.2 Distilling Pre-trained VLMs to LinMU

Although our proposed architecture achieves linear complexity, directly training a VLM with the new architecture from scratch would be very costly. We, therefore, propose a progressive distillation framework that converts a pretrained Transformer-based VLM (teacher) into LinMU (student) by replacing each quadratic self-attention layer with an M-MATE block of identical input/output size, as described in Alg. 2. Fig. 5 illustrates the overall replacement and the three-stage distillation schedule.

---

**Algorithm 1** M-MATE block forward

---

**Require:** tokens $\{\mathbf{z}_t\}_{t=1}^N$, vision token mask $\mathcal{M}_V$
 1: **Flex-MA branch**
 2: $\{\mathbf{u}_t\} \leftarrow \text{RMS}(\{\mathbf{z}_t\})$
 3: $\{\mathbf{v}_t^{\text{Flex}}\} \leftarrow \text{Flex-BiMamba2}(\{\mathbf{u}_t\})$
 4: $\{\mathbf{y}_t^{\text{Flex}}\} \leftarrow \text{Rev-RMS}(\{\mathbf{v}_t\})$
 5: **Local-Swin branch**
 6: $\{\mathbf{v}_t^{\text{LS}}\} \leftarrow \text{LocalSwin}(\mathcal{M}_V \odot \{\mathbf{z}_t\})$              ▷ alternately shifted fixed-size windows
 7: $\{\mathbf{y}_t^{\text{LS}}\} \leftarrow \text{Pad}(\{\mathbf{v_t^{\text{LS}}}\})$              ▷ padding from vision tokens to the full sequence
 8: **return** $\mathbf{y}_t = \text{LayerNorm}(\lambda_t \mathbf{y}_t^{\text{Flex}} + (1 - \lambda_t)\mathbf{y}_t^{\text{LS}})$              ▷ $\lambda_t$ is learnable

---

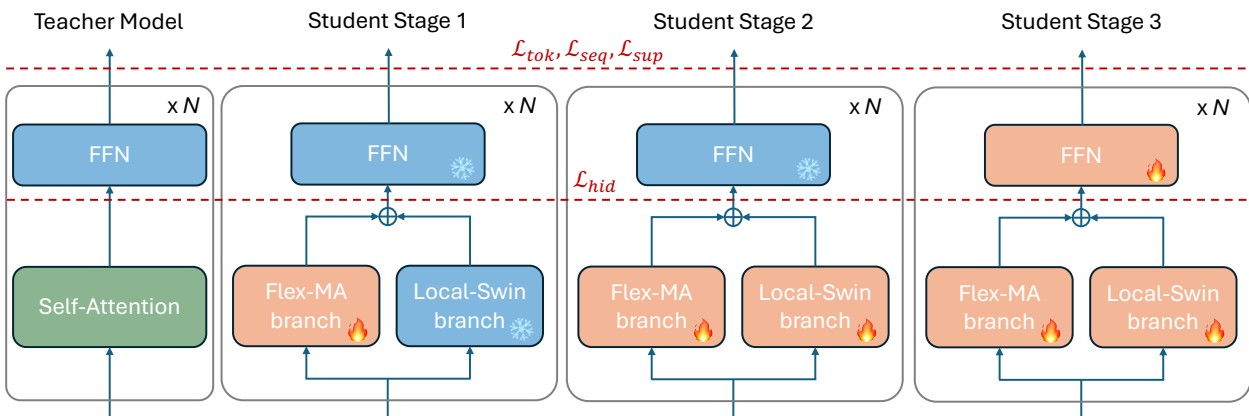

Figure 5: Our proposed three-stage distillation pipeline to replace attention layers in pre-trained VLMs with linear-complexity M-MATE blocks.

### 3.2.1 Weight Reuse Initialization

Simply swapping attention for randomly initialized M-MATE blocks degrades accuracy. Following the weight-reuse spirit in MambaInLlama (Wang et al., 2024a) and LinGen-Uni (Wang et al., 2025a), we initialize student M-MATE blocks from the teacher attention weights wherever possible.

Let the teacher attention layer $\ell$ have (flattened) projection matrices $W_Q^\ell, W_K^\ell, W_V^\ell \in \mathbb{R}^{d \times d}$ and output projection $W_O^\ell \in \mathbb{R}^{d \times d}$. For the Flex-MA (masked bidirectional Mamba2) branch, we reuse:

$$W_C^\ell \leftarrow W_Q^\ell, \quad W_B^\ell \leftarrow W_K^\ell, \quad W_X^\ell \leftarrow W_V^\ell, \quad W_{O,\text{Flex}}^\ell \leftarrow W_O^\ell, \tag{4}$$

where $(W_C^\ell, W_B^\ell, W_X^\ell)$ are the dominant linear projections in Mamba2 (accounting for over 90% parameters), while the state-transition parameters (e.g., $A$) use standard initialization and gates use identity initialization.

We initialize the Local-Swin branch as the teacher attention restricted to fixed local windows over vision tokens as follows:

$$(W_{Q,\text{Swin}}^\ell, W_{K,\text{Swin}}^\ell, W_{V,\text{Swin}}^\ell, W_{O,\text{Swin}}^\ell) \leftarrow (W_Q^\ell, W_K^\ell, W_V^\ell, W_O^\ell), \tag{5}$$

while enforcing the Swin window mask. Intuitively, some of the teacher's attention heads are likely specialized to local structure (as commonly observed in early Transformer layers); hence, the Local-Swin branch can inherit them. This weight reuse gives the student a head start, often starting in a regime not too far from the teacher's function.

---

**Algorithm 2** Progressive Distillation from a Transformer VLM to LinMU

---

**Require:** Frozen teacher $f_{\mathrm{T}}$ (self-attention); student $f_{\mathrm{S}}$ (M-MATE); dataset $\mathcal{D}$; temperature $\tau$; steps $(S_1, S_2, S_3)$ for each stage

**Require:** Loss weights $\lambda_{\mathrm{hid}}, \lambda_{\mathrm{tok}}, \lambda_{\mathrm{seq}}, \lambda_{\mathrm{sup}}$

**Ensure:** Distilled LinMU $f_{\mathrm{S}}$

 1: **Weight reuse init:** for each replaced layer $\ell$, apply Eq. 4 and Eq. 5; initialize remaining Mamba2 parameters in the standard way.
 2: Freeze all teacher params; freeze student backbone and vision encoder/projector.
 3: **Stage 1 (Flex-MA only):** unfreeze $\theta_{\mathrm{Flex}}$; disable/freeze Local-Swin.
 4: **for** $s = 1$ to $S_1$ **do**
 5:     Sample minibatch $(\mathbf{x}, \mathbf{y}^{\mathrm{gt}}) \sim \mathcal{D}$
 6:     $\{\mathbf{h}_\ell^{\mathrm{T}}\}, \{\mathbf{z}_{\mathrm{T},t}\} \leftarrow f_{\mathrm{T}}(\mathbf{x})$                                         ▷ no grad
 7:     $\{\mathbf{h}_\ell^{\mathrm{S}}\}, \{\mathbf{z}_{\mathrm{S},t}\} \leftarrow f_{\mathrm{S}}(\mathbf{x})$
 8:     Compute $\mathcal{L}_{\mathrm{hid}}$ by Eq. 6 and $\mathcal{L}_{\mathrm{tok}}$ by Eq. 7
 9:     $\mathcal{L}^{(1)} \leftarrow \lambda_{\mathrm{hid}}\mathcal{L}_{\mathrm{hid}} + \lambda_{\mathrm{tok}}\mathcal{L}_{\mathrm{tok}}$                              ▷ Eq. 10
10:     Update $\theta_{\mathrm{Flex}}$ with $\nabla\mathcal{L}^{(1)}$
11: **end for**
12: **Stage 2 (Flex-MA + Local-Swin):** unfreeze $\theta_{\mathrm{Swin}}$; keep backbone frozen.
13: **for** $s = 1$ to $S_2$ **do**
14:     Sample minibatch $(\mathbf{x}, \mathbf{y}^{\mathrm{gt}}) \sim \mathcal{D}$
15:     $\{\mathbf{h}_\ell^{\mathrm{T}}\}, \{\mathbf{z}_{\mathrm{T},t}\} \leftarrow f_{\mathrm{T}}(\mathbf{x})$                                         ▷ no grad
16:     $\{\mathbf{h}_\ell^{\mathrm{S}}\}, \{\mathbf{z}_{\mathrm{S},t}\} \leftarrow f_{\mathrm{S}}(\mathbf{x})$
17:     Compute $\mathcal{L}_{\mathrm{hid}}$ (Eq. 6), $\mathcal{L}_{\mathrm{tok}}$ (Eq. 7)
18:     $\mathcal{L}^{(2)} \leftarrow \lambda_{\mathrm{hid}}\mathcal{L}_{\mathrm{hid}} + \lambda_{\mathrm{tok}}\mathcal{L}_{\mathrm{tok}}$                              ▷ Eq. 11
19:     Update $(\theta_{\mathrm{Flex}}, \theta_{\mathrm{Swin}})$ with $\nabla\mathcal{L}^{(2)}$
20: **end for**
21: **Stage 3 (LoRA fine-tune):** insert LoRA in backbone; unfreeze $\theta_{\mathrm{LoRA}}$ and all M-MATE params.
22: **for** $s = 1$ to $S_3$ **do**
23:     Sample minibatch $(\mathbf{x}, \mathbf{y}^{\mathrm{gt}}) \sim \mathcal{D}$
24:     $\{\mathbf{z}_{\mathrm{T},t}\} \leftarrow f_{\mathrm{T}}(\mathbf{x})$ and $\tilde{\mathbf{y}} \leftarrow \mathrm{Decode}(f_{\mathrm{T}}, \mathbf{x})$                      ▷ no grad
25:     $\{\mathbf{z}_{\mathrm{S},t}\} \leftarrow f_{\mathrm{S}}(\mathbf{x})$
26:     Compute $\mathcal{L}_{\mathrm{tok}}$ (Eq. 7), $\mathcal{L}_{\mathrm{seq}}$ (Eq. 8), and $\mathcal{L}_{\mathrm{sup}}$ (if $\mathbf{y}^{\mathrm{gt}}$ available)
27:     $\mathcal{L}^{(3)} \leftarrow \lambda_{\mathrm{tok}}\mathcal{L}_{\mathrm{tok}} + \lambda_{\mathrm{seq}}\mathcal{L}_{\mathrm{seq}} + \lambda_{\mathrm{sup}}\mathcal{L}_{\mathrm{sup}}$                ▷ Eq. 13
28:     Update $(\theta_{\mathrm{M\text{-}MATE}}, \theta_{\mathrm{LoRA}})$ with $\nabla\mathcal{L}^{(3)}$
29: **end for**
30: **return** $f_{\mathrm{S}}$

---

### 3.2.2 Three-stage progressive distillation

We keep the teacher $f_{\mathrm{T}}$ frozen and train the student $f_{\mathrm{S}}$ via three stages that progressively increase trainable capacity. Let $\mathcal{A}$ denote the set of replaced layers.

**Common loss terms.** We use (i) *layer-wise main path matching* between the output of the M-MATE block and the replaced attention layer:

$$\mathcal{L}_{\mathrm{hid}} = \frac{1}{|\mathcal{A}|} \sum_{\ell \in \mathcal{A}} \left[ \frac{1}{N} \sum_{t=1}^{N} \left\| \mathbf{h}_{\ell,t}^{\mathrm{S}} - \mathbf{h}_{\ell,t}^{\mathrm{T}} \right\|_2^2 \right] \tag{6}$$

where $\{\mathbf{h}_{\ell,t}^{\mathrm{S}}\}_{t=1}^{N}$ and $\{\mathbf{h}_{\ell,t}^{\mathrm{T}}\}_{t=1}^{N}$ are the output of the student M-MATE block and the teacher attention module at the $\ell$-th layer, respectively. (ii) *token-level knowledge distillation (KD)* with temperature $\tau$:

$$\mathcal{L}_{\mathrm{tok}} = \frac{\tau^2}{T} \sum_{t=1}^{T} \mathrm{KL}\big(p_{\mathrm{T},t}^{\tau} \,\|\, p_{\mathrm{S},t}^{\tau}\big), \quad p_{\mathrm{T},t}^{\tau} = \mathrm{softmax}(\mathbf{z}_{\mathrm{T},t}/\tau), \; p_{\mathrm{S},t}^{\tau} = \mathrm{softmax}(\mathbf{z}_{\mathrm{S},t}/\tau), \tag{7}$$

where $\mathbf{z}_{\cdot,t}$ are vocabulary logits at generation step $t$ and $T$ is the output length. We use this loss to keep the student aligned on end-task predictions with the teacher model. (iii) *sequence-level KD* using a teacher-decoded pseudo target $\tilde{\mathbf{y}} = \text{Decode}(f_{\text{T}}, \mathbf{x})$:

$$\mathcal{L}_{\text{seq}} = -\frac{1}{T} \sum_{t=1}^{T} \log p_{\text{S}}(\hat{y}_t \mid \mathbf{x}, \tilde{\mathbf{y}}_{<t}) \tag{8}$$

where $\hat{y}_t$ is the output of the student model. This two-level KD ($\mathcal{L}_{\text{tok}}$ and $\mathcal{L}_{\text{seq}}$, which are often denoted as word-level and sequence-level loss, respectively) has been shown to be effective for distilling LLMs, as it balances imitating the probability distribution with matching the single best sequence. (iv) *ground-truth task loss* when the ground truth $\mathbf{y}^{\text{gt}}$ is available:

$$\mathcal{L}_{\text{sup}} = \mathcal{L}_{\text{task}}(\hat{y}, \mathbf{y}^{\text{gt}}) \tag{9}$$

**Stage 1: Flex-MA-only distillation.** We freeze the entire student except the Flex-MA branch parameters $\theta_{\text{Flex}}$ and freeze the Local-Swin branch. The objective emphasizes *module-level imitation* while retaining end-task alignment:

$$\min_{\theta_{\text{Flex}}} \mathcal{L}^{(1)} = \lambda_{\text{hid}} \mathcal{L}_{\text{hid}} + \lambda_{\text{tok}} \mathcal{L}_{\text{tok}} \tag{10}$$

Matching $\{\mathbf{h}_{\ell,t}^{\text{T}}\}_{t=1}^{N}$ forces the Flex-MA branch to approximate the teacher attention *effect* at each replaced layer, providing a strong and stable signal even when only a small subset of parameters is trainable.

**Stage 2: Joint distillation of Flex-MA and Local-Swin.** We unfreeze Local-Swin parameters $\theta_{\text{Swin}}$ and train both branches jointly while keeping the rest of the backbone frozen. We initialize the Local-Swin branch as aforementioned and allow it to learn to model the finer local details that the Flex-MA branch might have missed due to the adjacency preservation issue. The Stage-2 objective is:

$$\min_{\theta_{\text{Flex}}, \theta_{\text{Swin}}} \mathcal{L}^{(2)} = \lambda_{\text{hid}} \mathcal{L}_{\text{hid}} + \lambda_{\text{tok}} \mathcal{L}_{\text{tok}} \tag{11}$$

Because the Local-Swin branch starts from the teacher's weights and the Flex-MA branch has been trained in Stage 1, the joint optimization in Stage 2 converges quickly. We find that if we attempt to train both branches from the start, the Flex-MA branch is less stable and harder to converge; by staging it, each branch learns its role: Flex-MA first learns global context, then Local-Swin fills in local consistency. By the end of Stage 2, the student's M-MATE blocks can produce outputs very close to the teacher's attention outputs for each layer.

**Stage 3: LoRA fine-tuning of the backbone.** In the final stage, we unfreeze the rest of the student model (the vision encoder and the corresponding projector are still fixed). However, instead of full fine-tuning (which could be too costly and risk distorting the pretrained weights), we apply low-rank adaptation (LoRA) (Hu et al., 2022) to these layers. For a frozen weight $W \in \mathbb{R}^{d_{\text{out}} \times d_{\text{in}}}$, LoRA parameterizes

$$W' = W + \Delta W, \qquad \Delta W = BA, \quad A \in \mathbb{R}^{r \times d_{\text{in}}}, \ B \in \mathbb{R}^{d_{\text{out}} \times r}, \tag{12}$$

and only $(A, B)$ are trainable. We train all M-MATE parameters plus LoRA parameters $\theta_{\text{LoRA}}$ of FFNs in the backbone with an objective that shifts emphasis to output-level alignment and (optionally) ground-truth supervision:

$$\min_{\theta_{\text{M-MATE}}, \theta_{\text{LoRA}}} \mathcal{L}^{(3)} = \lambda_{\text{tok}} \mathcal{L}_{\text{tok}} + \lambda_{\text{seq}} \mathcal{L}_{\text{seq}} + \lambda_{\text{sup}} \mathcal{L}_{\text{sup}} \tag{13}$$

Note that at this stage, we incorporate $\mathcal{L}_{\text{seq}}$ and $\mathcal{L}_{\text{sup}}$ for better output distribution alignment but exclude $\mathcal{L}_{\text{hid}}$ because other blocks in the student backbone are also being trained. This stage yields the final LinMU model. Notably, because LoRA updates are lightweight, the original knowledge in the backbone is largely preserved.

We set $\lambda_{\mathrm{hid}} = \lambda_{\mathrm{tok}} = \lambda_{\mathrm{seq}} = \lambda_{\mathrm{sup}} = 0.5$ in all the three stages mentioned above. In summary, weight reuse provides a strong functional prior, while the three-stage schedule stabilizes optimization: Flex-MA first learns to reproduce long-range/contextual effects, Local-Swin then specializes to local visual structure, and LoRA fine-tuning adapts the remaining backbone to the new linear-complexity blocks with minimal parameter updates. Through this distillation pipeline, LinMU learns to faithfully emulate the original model's attention-driven reasoning, but using only linear-complexity blocks. The outcome is a model that, during inference, requires no quadratic attention computation and is, therefore, much more efficient on long sequences (e.g., dealing with high-resolution images or long videos).

## 4 Experiments

In this section, we first introduce the general experimental setup in Sec. 4.1, including backbones, datasets, and benchmarks that we use to deploy our framework, perform distillation, and evaluate performance. Then in Sec. 4.2, we compare the performance and efficiency of our proposed framework with baseline models, demonstrating that LinMU achieves a better balance between accuracy and efficiency. Finally, we provide ablation experimental results to validate the effectiveness of the techniques proposed and design choices made in our framework in Sec 4.3.

### 4.1 Experimental Setup

**Backbones.** For most experiments, we use NVILA-Video-8B as the teacher model to deploy our proposed distillation framework and achieve linear complexity. It is an 8B-parameter VLM that exhibits competitive performance on many image and video understanding benchmarks. For image benchmarks, we use NVILA-8B as the teacher model instead; it is a checkpoint before the video-instruction tuning stage and focuses more on image tasks. To demonstrate the generalization ability of LinMU, we further perform distillation on Qwen2.5-VL-7B-Instruct (Bai et al., 2025), which has strong performance across various perception tasks, including but not limited to long document understanding, video grounding, and OCR.

**Datasets.** Referring to the image and video instruction-tuning datasets of NVILA, we involve (1) the single-image and multi-image subsets of OneVision-1.6M in LLaVA-OneVision (Li et al., 2024), which include around 1M samples; (2) LLaVa-Video-178K (Zhang et al., 2024b), which provides around 178K videos; (3) the training set of ActivityNet-QA (excluding the test set), which includes around 60K question-answer (QA) pairs on 6K videos as our distillation dataset. ActivityNet-QA is only involved in Stage 3. Note that Qwen2.5-VL-7B-Instruct performs well across many downstream tasks, including long document understanding. Complete distillation of it requires building a comprehensive dataset including all its capabilities, while our datasets here focus more on the domain that we target (i.e., multimodal understanding). Distilling Qwen2.5-VL-7B-Instruct is a proof of concept that our proposed framework can be deployed on various VLM backbones with a relatively standard model architecture design.

**Benchmarks.** We evaluate LinMU and baselines on a suite of image and video understanding tasks to verify that LinMU achieves comparable performance to the teacher model while gaining efficiency. (1) MMMU (Massive Multi-discipline Multimodal Understanding and Reasoning): A multimodal reasoning benchmark (Yue et al., 2024) spanning many academic subjects (e.g., science, math, engineering), where models answer questions that require understanding images like diagrams, charts, tables, and figures alongside text. (2) TextVQA: A visual QA benchmark (Singh et al., 2019) focused on reading and reasoning about text in images (scene text), not just recognizing objects. (3) ActivityNet-QA: A video QA benchmark (Yu et al., 2019) built on ActivityNet videos, testing whether a model can answer natural-language questions about actions/events in videos, often requiring temporal understanding. (4) LongVideoBench: A benchmark (Wu et al., 2024) designed for long-form video understanding, emphasizing long-context temporal reasoning (tracking events across minutes-long videos rather than short clips). (5) MLVU (Multi-task Long Video Understanding: A multi-task long-video understanding benchmark (Zhou et al., 2025) that evaluates a model's ability to comprehend lengthy videos across tasks like QA and other video-language understanding settings, stressing temporal reasoning over extended context. (6) Video-MME (Video Multi-Modal Evaluation): A comprehensive evaluation suite (Fu et al., 2025) for video-

Table 1: Performance comparison between the student LinMU model, the teacher NVILA model, and other baseline models. Note that GPT-4o is a strong closed-source baseline for reference.

| Models | Size | MMMU test | MMMU pro | TextVQA val | ActivityNet-QA acc. | ActivityNet-QA score | LongVideoBench val | LongVideoBench test | MLVU m-avg | Video-MME w/o sub. | Video-MME w/sub. |
|---|---|---|---|---|---|---|---|---|---|---|---|
| | | | | | *Transformer-based VLMs* | | | | | | |
| GPT-4o | — | 64.7 | 51.9 | 77.4 | 61.9 | — | 66.7 | 66.7 | 64.6 | 71.9 | 77.2 |
| LLaVA-OV | 7B | 42.8 | 24.1 | 78.3 | 56.6 | — | 56.5 | — | 64.7 | 58.2 | 61.5 |
| Qwen2-VL | 8B | 46.6 | 30.5 | 84.3 | — | — | 55.6 | 56.8 | 65.5 | 63.3 | 69.0 |
| InternVL2 | 8B | 42.6 | 29.0 | 77.4 | — | — | 54.6 | — | 64.0 | 56.3 | 59.3 |
| | | | | | *Linear-Complexity or Quadratic-Linear-Hybrid VLMs* | | | | | | |
| VL-Mamba | 2.8B | — | — | 48.9 | — | — | — | — | — | — | — |
| ViRWKV-HM | 8.7B | — | — | 56.7 | — | — | — | — | — | — | — |
| LongLLaVA | 9B | 34.4 | — | — | — | — | 51.9 | — | 53.3 | 51.6 | — |
| LongVU | 7B | — | — | — | — | — | 53.5 | — | 65.4 | 60.6 | — |
| VAMBA | 10B | — | — | — | — | — | 55.9 | — | 65.9 | 57.8 | — |
| | | | | | *Teacher and Student Models* | | | | | | |
| NVILA | 8B | 44.4 | **27.8** | **80.1** | **60.9** | **3.7** | **57.7** | 58.7 | **70.1** | 64.2 | 70.0 |
| LinMU-NV | 8B | **44.6** | 27.3 | 79.3 | 60.1 | 3.6 | 57.4 | **58.8** | 69.4 | **64.5** | **70.1** |

language models that uses standardized tests (often multiple-choice) to measure general video understanding across diverse categories and video lengths.

**Training recipe.** The distillation training is done on $8\times$A100 GPUs, using the AdamW optimizer. In Stages 1 and 2, we use a Learning Rate (LR) of 5e-4 for M-MATE parameters (teacher backbone frozen). In Stage 3, we apply LoRA (rank=8) to the backbone and use a smaller LR (2e-5). We stop Stage 1 after 10K steps (when the Flex-MA branch's hidden alignment loss plateaus), Stage 2 after 15K more steps, and Stage 3 after 25K steps. The total wall time for distilling NVILA-Video-8B is about 50 hours. We use the same training recipe for Qwen2.5-VL-7B-Instruct.

## 4.2 Better Balance between Accuracy and Efficiency

**Performance compared to teacher**: Table 1 summarizes the performance of the student LinMU model against the teacher NVILA model on image and video benchmarks. We also include GPT-4o (Hurst et al., 2024) as a strong closed-source baseline, as well as other typical open-source baselines, including LLaVA-OneVision (Li et al., 2024), Qwen2-VL (Wang et al., 2024b), and InternVL2 (Chen et al., 2024d), and linear-complexity or quadratic-linear-hybrid baselines, such as VL-Mamba (Qiao et al., 2024), VisualRWKV-HM (Hou et al., 2025a), LongLLaVA (Wang et al., 2024c), LongVU (Shen et al., 2024), and VAMBA (Ren et al., 2025), for reference. LinMU achieves performance that is very close to the teacher across the board, even outperforming the teacher model on a few benchmarks.

For example, on ActivityNet-QA, LinMU reaches 60.1% accuracy, essentially matching NVILA's 60.9%. On TextVQA, LinMU scores 79.3%, only slightly below the teacher's 80.1%. On the test sets of MMMU (44.6% vs. 44.4%) and LongVideoBench (58.8% vs. 58.7%), and across both settings of Video-MME (64.5% vs. 64.2%; 70.1% vs. 70.0%), LinMU achieves slightly better performance than NVILA. Notably, LinMU outperforms many other models of similar scale on these benchmarks, benefiting from the strong performance of the teacher model. For instance, it outperforms the prior 7B model, LLaVA-OneVision, and the 8B model, InternVL2. It is also on par with the recent Qwen2-VL model.

MMMU and Video-MME tests complex multi-step reasoning; MLVU and LongVideoBench specifically target long video understanding, where the advantage of linear complexity is expected to be most useful. After distillation, LinMU remains competitive on these benchmarks, while achieving linear complexity without quadratic attention. This indicates that our distillation successfully transfers the needed capability from the teacher to the linear student.

**Inference efficiency**: We measure the efficiency of VLMs in terms of TTFT and token throughput. TTFT reflects the time required to generate the first output token after the VLM receives the input, and token

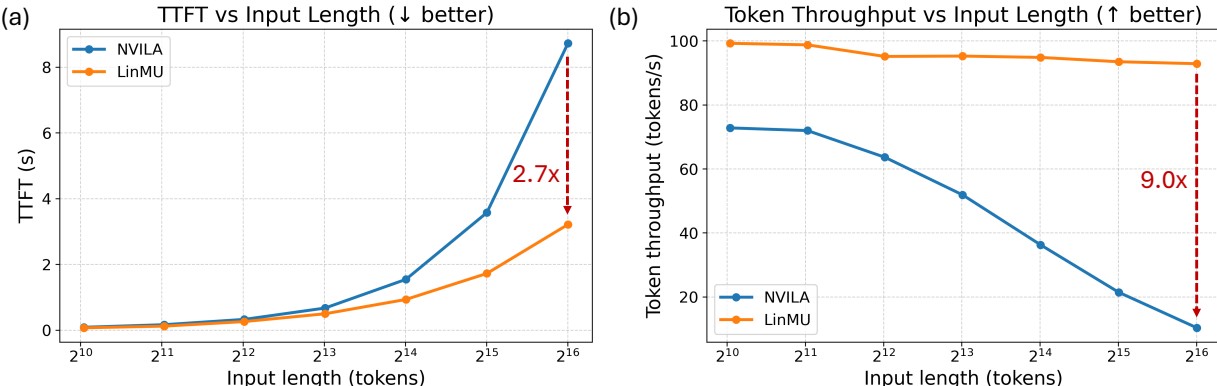

Figure 6: Efficiency comparison between the quadratic-complexity teacher model NVILA and the linear-complexity student model LinMU: how (a) TTFT (lower is better) and (b) token throughput (higher is better) scales as the input length increases. $2^{16}$ input tokens correspond to an 85-second 24fps video at 224×224 resolution with 14×14 spatial compression rate and 8× temporal compression rate.

throughput describes the rate at which output tokens are generated. Note that standard attention with modern efficient kernels, such as FlashAttention, already has a linear memory footprint with respect to sequence length. Thus, replacing these attention layers with M-MATE blocks does not yield significant memory savings. Thus, we focus on the latency reduction and throughput improvements they provide.

The scaling curves of TTFT and token throughput with respect to the input sequence length are shown in Fig. 6. Results are measured on NVIDIA H100 GPUs. Note that we exclude the vision encoding time here to highlight the LM Decoder cost that we are most interested in. Longer input sequences correspond to higher-resolution images or longer videos. $2^{16}$ input tokens correspond to an 85-second 24fps video (2048 frames) at 224×224 resolution with 14×14 spatial compression rate and 8× temporal compression rate. Fig. 6 validates the complexity gain of LinMU over the standard attention-based VLM architecture. The TTFT of LinMU scales much more slowly than that of NVILA as the input length increases, achieving 2.7× speed-up when dealing with minute-length videos, even at low resolution. The speed-up of LinMU in terms of token throughput is more significant: 9× higher than that of NVILA when dealing with a minute-length video at 224×224 resolution. LinMU maintains almost the same token throughput as the input sequence length increases, whereas NVILA's token throughput drops significantly. Benefiting from the linear-complexity of LinMU, the longer the input sequence is, the more speed-up LinMU can achieve. The efficiency gains imply LinMU can enable longer-context reasoning (e.g., analyze an entire movie or an hour-long surveillance video continuously) and can be deployed in scenarios where NVILA would be too slow (e.g., real-time robot perception with limited compute). Note that LinMU achieves these gains while maintaining competitive performance relative to NVILA across various tasks.

### 4.3 Ablation Studies

We conduct ablations to support our design choices in the distillation process, justify the M-MATE block compositions, and validate the universality of our proposed LinMU framework.

**Effect of distillation stages.** We record performance improvement across the three distillation stages to validate their effectiveness in Table 2. Even with only initialization, LinMU can finish some simple understanding tasks. As distillation progresses, its performance across various tasks gradually improves. In particular, Stage 1 forces the masked bidirectional Mamba2 to approximate global attention patterns, while Stage 2 trains the model to fit local patterns using the Local-Swin branch. Stage 3 allows the backbone to adapt to and calibrate the interactions between M-MATE blocks and the rest of the model. If we train LinMU in a single stage (i.e., unfreeze both branches and the backbone from scratch, and train the student

Table 2: Performance comparison between models at different training status. The performance of LinMU gradually improves as the distillation proceeds. Finishing distillation in a single stage leads to a lower performance under the same compute budget.

| Training Status | ActivityNet-QA$_{acc.}$ | LongVideoBench$_{test}$ | MLVU | Video-MME$_{w/sub.}$ |
|---|---|---|---|---|
| Initialization | 39.3 | 29.4 | 48.7 | 43.2 |
| Stage 1 | 48.7 | 38.5 | 58.3 | 52.9 |
| Stage 1+2 | 56.7 | 52.7 | 65.2 | 63.4 |
| Stage 1+2+3 | **60.1** | **58.8** | **69.4** | **70.1** |
| All in one stage | 57.3 | 54.2 | 66.8 | 67.0 |

Table 3: Performance comparison between different distillation loss settings. Reducing the corresponding factor to zero means eliminating that loss. The default setting (in the first row) achieves the best balanced performance across various tasks.

| $\lambda_{hid}$ | $\lambda_{seq}$ | $\lambda_{sup}$ | ActivityNet-QA$_{acc.}$ | LongVideoBench$_{test}$ | MLVU | Video-MME$_{w/sub.}$ |
|---|---|---|---|---|---|---|
| 0.5 | 0.5 | 0.5 | 60.1 | 58.8 | **69.4** | **70.1** |
| 0.5 | 0.5 | 0 | 58.7 | 58.6 | 69.1 | 69.8 |
| 0.5 | 0 | 0.5 | 59.2 | 57.1 | 68.5 | 68.2 |
| 0 | 0.5 | 0.5 | 58.1 | 56.6 | 67.9 | 68.6 |
| 0.5 | 0.5 | 1 | **60.8** | 56.9 | 68.2 | 69.3 |
| 0.5 | 1 | 0.5 | 59.4 | 58.2 | 68.8 | 69.5 |
| 1 | 0.5 | 0.5 | 59.9 | **58.9** | 69.1 | 69.4 |

model end-to-end), the student converges much more slowly and achieves lower final performance. This indicates that the stage-wise distillation is important.

**Loss function choices.** We conduct distillation experiments with different loss function settings to investigate the effectiveness of each loss term and compare their performance in Table 3. Note that the token-level KD loss $\mathcal{L}_{tok}$ is the standard distillation loss; hence, it is always retained with $\lambda_{tok} = 0.5$. Results presented in Table 3 indicate that eliminating any of the hidden feature loss $\mathcal{L}_{hid}$, sequence-level loss $\mathcal{L}_{seq}$, and ground-truth task loss $\mathcal{L}_{sup}$ incurs performance degradation. Without the hidden feature loss $\mathcal{L}_{hid}$, the student model converges more slowly and reaches a lower final performance under the same compute budget. Removing the sequence-level loss $\mathcal{L}_{seq}$ makes it harder for the student model to mimic the teacher model's behavior. The ground-truth task loss $\mathcal{L}_{sup}$ is especially helpful for improving the student performance on well-defined tasks with ground-truth labels, such as ActivityNet-QA. Our combined loss gives the best results, echoing findings from prior distillation research. We further explore increasing the factors of these loss terms from 0.5 to 1. Although increasing $\lambda_{sup}$ improves the student performance on ActivityNet-QA, the performance on other tasks suffers. Overall, our default setting (i.e., the first row in Table 3) achieves the best balance among various tasks.

**Window size of the Local-Swin branch.** We conduct a distillation ablation on the Local-Swin branch with different spatial and temporal window sizes, and report the results in Table 4. As shown, decreasing the window size along either dimension yields only marginal speedups, since the Local-Swin branch is already highly efficient. In contrast, enlarging the window spatially or temporally substantially increases latency while providing only limited accuracy gains. This is expected: the Local-Swin branch is primarily responsible for capturing short-range and localized correlations, whereas medium-range dependencies are effectively modeled by the Flex-MA branch.

**M-MATE block components.** We validate the effectiveness and cost of the Flex-MA branch and the Local-Swin branch in each M-MATE block in Table 5. If we drop the Local-Swin branch entirely (making LinMU a pure Mamba2-based model), performance degrades: 51.3% on LongVideoBench (vs. 58.8% for full LinMU) and 62.7% on Video-MME (vs. 70.1% for full LinMU). This is expected, as a pure SSM struggles with fine local details in vision due to the adjacency preservation issue. The Local-Swin branch addresses this issue

Table 4: Performance and efficiency comparison of different window sizes of the Local-Swin branch. We only compare their performance after Stage 2 to save computational resources. TTFT is measured at 32K input sequence length.

| Window Size | ActivityNet-QA$_{acc.}$ | LongVideoBench$_{test}$ | MLVU | Video-MME$_{w/sub.}$ | TTFT |
|---|---|---|---|---|---|
| 32×4×4 | 56.6 | **52.9** | 65.2 | **63.4** | 2.043 |
| 16×8×8 | **56.8** | 52.6 | **65.3** | 63.2 | 2.421 |
| 16×4×4 | 56.7 | 52.7 | 65.2 | **63.4** | 1.723 |
| 16×2×2 | 56.5 | 52.8 | 64.8 | 63.3 | **1.689** |
| 8×4×4 | 56.7 | 52.2 | 64.6 | 62.9 | 1.692 |

Table 5: Performance and efficiency comparison between different M-MATE branch settings. TTFT (s) and token throughput (tokens/s) are measured at 32K input sequence length.

| Flex-MA | Local-Swin | LongVideoBench$_{test}$ | MLVU | Video-MME$_{w/sub.}$ | TTFT | token throughput |
|---|---|---|---|---|---|---|
| Yes | Yes | **58.8** | **69.4** | **70.1** | 1.723 | 93.46 |
| Yes | No | 51.3 | 62.9 | 62.7 | 1.549 | 101.48 |
| No | Yes | 30.2 | 45.4 | 36.1 | 0.987 | 133.43 |

and improves the model performance on dealing with complex vision tasks (e.g., long video understanding) noticeably. Note that the Local-Swin branch maintains the linear complexity with respect to the number of tokens and only introduces tiny extra latency (e.g., 10% TTFT increment). On the contrary, removing the Flex-MA branch reduces TTFT and boosts token throughput significantly. However, this makes LinMU a pure window-attention-based model with a fixed small window size. As shown in Table 5, in this case, the model performance degrades significantly on all benchmarks. This indicates that it is infeasible to build a powerful VLM solely using window attention with small window sizes. These ablations confirm that the dual-branch design of M-MATE is critical for success: The Local-Swin branch addresses adjacent correlations, and the Flex-MA branch addresses medium-range and long-range correlations efficiently.

**VLM backbone besides NVILA.** To validate the generalization ability of our proposed framework, we further perform distillation on Qwen2.5-VL-7B-Instruct and transform it into the LinMU architecture to achieve linear complexity. Performance comparison between the teacher and student models is shown in Table 6. LinMU-Qwen still maintains the performance of the teacher model across various benchmarks. We further benchmark the efficiency gain brought about by LinMU in Fig. 7. It illustrates that the TTFT and token throughput of LinMU-Qwen scale much more slowly than the teacher model as the number of video frames (i.e., the input sequence length) increases. LinMU achieves 2.3× (4.9×) speed-up in terms of TTFT (token throughput) compared to the teacher model for 512-frame 256px videos. As the input video becomes longer or higher-resolution, LinMU delivers larger speedups.

## 4.4 Training Cost Analysis

Our total distillation and fine-tuning cost is about 400 A100 hours, which approximately equals 5.6 H100 GPU days. This is significantly lower than the cost of training a VLM from scratch. Specifically, NVILA initializes its vision encoder from SigCLIP and its LM decoder from Qwen2, but still uses 208 H100 GPU days to complete training. Qwen2.5-VL-7B initializes its LM decoder with Qwen2.5 LLMs and still uses about 4.1T tokens to train its LM decoder, which requires at least thousands of H100 GPU days. The proposed distillation pipeline not only saves the training cost by 100-1000× but also outperforms the teacher model on some benchmarks, such as Video-MME, as shown in Table 1 and Table 6.

## 5 Limitation

In this article, we focus on replacing the self-attention layers of the LM decoder with our proposed M-MATE blocks, while leaving the vision encoder unchanged. The vision encoder also possibly involves global self-attention layers, which have quadratic complexity with respect to the number of patches or tokens. Based

Table 6: Performance comparison between the teacher Qwen2.5-VL-7B-Instruct and the student LinMU model. After distillation, LinMU maintains the good performance of Qwen on most tasks.

| Models | Size | MMMU pro | TextVQA val | LongVideoBench val | test | MLVU m-avg | Video-MME w/o sub. | w/sub. |
|---|---|---|---|---|---|---|---|---|
| Qwen2.5-VL | 7B | **38.3** | **84.9** | **56.0** | **57.3** | **70.2** | 65.1 | 71.6 |
| LinMU-Qwen | 7B | 37.3 | 83.2 | 55.4 | 56.5 | 69.6 | **65.2** | **71.8** |

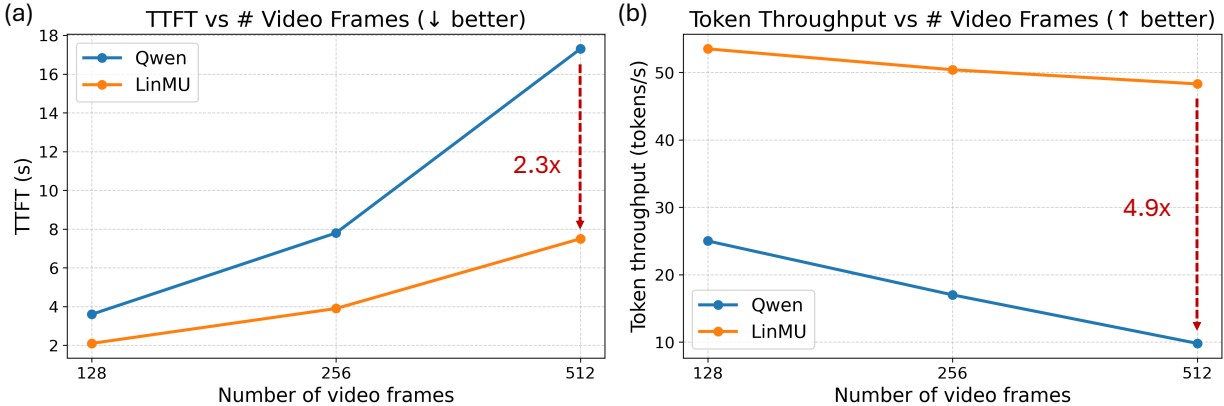

Figure 7: Efficiency comparison between the quadratic-complexity teacher model Qwen2.5-VL-7B-Instruct and the linear-complexity student model LinMU: how (a) TTFT (lower is better) and (b) token throughput (higher is better) scales as the number of input video frames. The metrics are measured on the captioning task for 224×224 videos. The corresponding input sequence length for 128, 256, and 512 video frames is 16K, 32K, and 64K.

on the current results on the LM decoder, we speculate that we can also replace the global self-attention layers in the vision encoder with our proposed M-MATE blocks and distillation pipeline, and we leave it as part of future work. However, it is also worth noting that: (1) The complexity of the vision encoder only affects TTFT. The token throughput, which is usually more important for rapid interactions, is solely determined by the complexity of the LM decoder. (2) The latency of the vision encoder in many modern VLMs is already almost linear with respect to the number of patches, benefiting from efficient attention kernels like FlashAttention (Dao et al., 2022). Specifically, the vision encoder of Qwen2.5-VL has 31 layers with window attention and only one layer with full attention. We also measured the latency of the vision encoder in NVILA under different numbers of input video frames, as shown in Table 7. The latency is almost linear with respect to the number of frames (i.e., the number of patches).

## 6 Conclusion

We introduced LinMU, a multimodal understanding model that achieves linear computational complexity in sequence length by distilling Transformer self-attention layers into M-MATE blocks. LinMU demonstrates that we can maintain high accuracy on challenging image-language and video-language tasks while eliminating the quadratic overhead of attention. Through a carefully crafted distillation process, which includes reusing pretrained weights, progressively training branches, and aligning the student with the teacher's behavior at multiple levels, LinMU effectively learns to replace attention with a combination of an efficient SSM and localized attention. Our experiments demonstrate that LinMU matches the performance of state-of-the-art teacher VLMs (including NVILA-8B-Video and Qwen2.5-VL-7B-Instruct) on multiple benchmarks, and substantially improves inference efficiency and scalability. These results are an encouraging sign that Transformer-quality multimodal reasoning can be achieved with more computationally scalable architectures.

Table 7: TTFT of NVILA contributed by the vision encoder under different numbers of video frames. The resolution is 224×224, and latencies are measured on a single H100 GPU.

| Number of Video Frames | 32 | 64 | 128 | 256 |
|---|---|---|---|---|
| Input Length (tokens) | 1K | 2K | 4K | 8K |
| Encoding Latency (s) | 0.2694 | 0.5372 | 1.068 | 2.137 |

Moving forward, this work opens up several exciting new directions. An immediate next step is to explore combining LinMU with token compression or adaptive input selection to further improve efficiency. Another direction is to extend linear multimodal modeling to native generative tasks, such as auto-regressive image generation driven by a VLM. In addition, while we distilled from 7-8B teacher models, this framework could also be applied to larger teachers (e.g., 33B) to train an efficient yet powerful student, or even to specialist models in medical or robotics domains where long video analysis is needed. Finally, as linear architectures mature, an important goal is to develop training paradigms that optimize linear models from scratch while maintaining competitive accuracy. LinMU's distillation results indicate that much of the self-attention representational capacity can be transferred to a linear form; a longer-term objective is to design dedicated training schemes (rather than directly inheriting Transformer recipes) that achieve this without relying on a Transformer teacher. We hope our work motivates broader exploration of attention-free or attention-lite multimodal models, making high-performing VLMs more accessible, faster, and better suited to ever-growing context lengths in real-world applications.

**Acknowledgments**

This work was supported in part by Qualcomm and in part by NSF under Grant No. CCF-2203399. We also appreciate the assistance provided by Hongxu Yin from NVIDIA Research.

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
