# OpenReview forum: "LinMU: Multimodal Understanding Made Linear"
_TMLR — Accepted by TMLR_

### Review · Reviewer_ciny · 2026-02-02

**Summary Of Contributions:**

Core Contributions:
1. Proposes LinMU, a VLM with pure linear complexity by replacing self-attention with the dual-branch M-MATE block (Flex-MA for global dependencies, Local-Swin for local correlations).
2. Develops a three-stage distillation framework (initialize with teacher weights, train Flex-MA, joint train both branches, LoRA fine-tune) to preserve teacher performance.
3. Matches state-of-the-art VLM performance on multiple benchmark, while boosting TTFT by 2.7× and token throughput by 9.0× for long videos.

Strengths:
1. Linear Complexity with Competitive Performance: The primary strength is achieving true linear complexity with respect to sequence length without the significant performance drop typically associated with non-attention architectures. It matches the accuracy of strong teacher models like NVILA and Qwen2.5-VL.
2. Effective Hybrid Design (M-MATE): The dual-branch design combines the strengths of SSMs (global mixing, linear cost) and window attention (local precision), mitigating the adjacency preservation issue that plagues flattened 1D sequence models in vision tasks.
3. Robust Distillation Strategy: The proposed three-stage distillation pipeline, combined with weight reuse from the teacher, ensures stable convergence and effective knowledge transfer, avoiding the high cost of pre-training a linear VLM from scratch.
4. Significant Efficiency Gains: The empirical results demonstrate substantial improvements in inference speed and throughput, particularly for long-context scenarios, making the model suitable for edge deployment or resource-constrained settings.

Weakness:
1. Reliance on Teacher Models: The framework is fundamentally a distillation method. It requires a high-quality, pre-trained quadratic VLM teacher to function. It does not explore training a linear VLM from scratch, meaning its performance is likely capped by the teacher's capabilities.
2. Complexity of Implementation: Compared to a simple drop-in replacement, the three-stage training pipeline with specific freezing schedules and loss combinations is relatively complex to implement and tune.
3. Limited to Decoder Modification: The method currently focuses on replacing attention in the Language Model decoder. The Vision Encoder remains unchanged, which might still be a bottleneck in fully end-to-end processing for extremely high-resolution inputs.
4. Fixed Window Limitations: While the Local-Swin branch is efficient, using a fixed small window size might still limit the modeling of medium-range dependencies that fall outside the window but are too subtle for the global Mamba branch to capture perfectly.

**Audience:**

Yes

**Audience Explanation:**

I believe the findings would be of significant interest to TMLR's audience.

1. This paper addresses fundamental scalability bottleneckss: the quadratic complexity of self-attention. As multimodal models increasingly handle high-resolution images and long-form video, computational scaling becomes a primary research concern.
2. This paper provide a novel integration of SSMs and Attention: the M-MATE block combines SSMs (Mamba2) with localized 3D Swin attention in a way that eliminates quadratic operations entirely.

**Broader Impact Concerns:**

This article does not include a section on Broader Impact, and I do not believe there is any need to discuss any Broader Impact.

**Claims And Evidence:**

Yes

**Claims Explanation:**

The evidence is accurate and clear to support the claims.

The claims of Linear Complexity is supported by theoretical grounding on section 3.1 and empirical scaling.
The claims of Performance Parity is supported by multi-benchmark validation.
The claims of Architectural Necessity is supported by clear ablation study.

**Requested Changes:**

1. Comparison with Existing Linear/Efficient VLMs
Table 1 compares LinMU primarily against standard, quadratic Transformer-based VLMs (NVILA, Qwen2-VL, LLaVA). While it is impressive that LinMU matches their performance, the paper lacks a direct performance comparison with other existing linear or efficient architectures mentioned in the Related Work, such as VL-Mamba, Cobra, or VisualRWKV.

2. Extend Ablation on Window Size in Local-Swin Branch
The paper uses a fixed 16×4×4 window size but does not justify its choice. Add ablation studies testing different window sizes (e.g., 8×4×4, 32×4×4) to show how local window scale impacts performance-efficiency tradeoffs. This is critical to validating the Local-Swin branch’s design rationale.

3. Complexity Analysis of the Vision Encoder
The paper’s title and abstract claim "Multimodal Understanding Made Linear." However, Section 3.1 explicitly states, "we leave the Vision Encoder unchanged." Many state-of-the-art VLMs utilize ViTs that possess quadratic complexity with respect to the number of patches per image/frame, or the number of frames if joint space-time attention is used.  If the Vision Encoder becomes the bottleneck at very long context lengths due to quadratic scaling, this limitation must be acknowledged.

---

> ### Author Response · Authors · 2026-03-02
>
> We thank the insightful comments from reviewer ciny. Regarding the mentioned weaknesses and requested changes, we respectfully respond as follows:
> 1. About the reliance on teacher models. On the one hand, our linear student can outperform the teacher model on some benchmarks, such as Video-MME, as shown in Table 1 of our paper. It is not capped by the teacher’s capabilities. On the other hand, training a powerful VLM from scratch is extremely expensive and not affordable for most academic researchers. NVILA initializes its vision encoder from SigLIP and its LM decoder from Qwen2, but it still uses 208 H100 GPU days to complete training. Qwen2.5-VL-7B uses about 4.1T tokens to train its LM decoder, which is initialized by Qwen2.5 LLM and requires at least thousands of H100 GPU days. For comparison, our fine-tuning cost is only 400 A100 hours, which approximately equals 5.6 H100 GPU days.
>
> 2. About the complexity of our proposed method. Our proposed method reuses more than 90% weights from the pre-trained teacher model for initialization and then gradually unfreezes more and more modules in the student model to finish distillation. It is motivated by a clear logic flow and is relatively easy to implement.
>
> 3. About the complexity of the Vision Encoder. We admit that the Vision Encoder is left unchanged and possibly involves some full-attention layers, which have quadratic complexity. We added a “Limitation” section to discuss this. Based on the current results on the LM decoder, we speculate that we can also replace the global self-attention layers in the vision encoder with our proposed M-MATE blocks and distillation pipeline, and we leave it as part of future work. It is worth noting that: (1) The complexity of the vision encoder only affects TTFT, and the token throughput, which is usually more important for rapid interactions, is solely determined by the complexity of the LM decoder. (2) The latency of the vision encoder in lots of modern VLMs is already almost linear with respect to the number of patches, benefiting from efficient attention kernels like FlashAttention. Specifically, the vision encoder of Qwen2.5-VL has 31 layers with window attention and only one layer with full attention. We also measured the latency of the vision encoder in NVILA under different numbers of input video frames, as shown in Table 7. The latency is also almost linear with respect to the number of frames (i.e., the number of patches).
>
> 4. About the fixed window. We believe the strong performance of student models on long video benchmarks, such as LongVideoBench, MLVU, and Video-MME, has proved that Mamba modules are actually very good at capturing medium-range dependencies, and the Local-Swin branch only needs to deal with very localized correlations.
>
> 5. About the comparison with other linear-complexity baselines. Lots of existing linear-complexity works didn’t train their model on video understanding tasks and are not large enough to perform fair comparisons (e.g., the largest size is a few hundred millions of parameters). To the best of our knowledge, we collected some linear-complexity and quadratic-linear-hybrid VLMs as the baselines and added them to Table 1. At similar or larger sizes, their performance lags behind NVILA and our LinMU student.
>
> 6. About the ablation on the window size of the Local-Swin branch. We supplemented the related ablation results in Section 4.3 “Ablation Studies”. The results indicate that smaller window sizes do not bring noticeable efficiency gains (because the Local-Swin branch is already super fast) and larger window sizes do not bring significant performance gains (because the Local-Swin branch only needs to deal with very localized correlations).

---

### Review · Reviewer_7owK · 2026-02-14

**Summary Of Contributions:**

The paper proposes LinMU, an efficient Vision-Language Model that converts existing quadratic-complexity self-attention layers into linear-complexity modules. The core design of LinMU lies in the M-MATE block, a drop-in replacement for attention that uses a dual-branch architecture: the Flex-MA branch captures global context efficiently with Mamba2, while the Local-Swin branch handles precise local correlations. On top of that, the paper introduces a three-stage knowledge distillation pipeline that effectively transfers knowledge from a pre-trained teacher VLM to the new LinMU student architecture.

**Additional Comments:**

- While the entire LinMU framework is effective, it is parasitic; it heavily relies on a powerful, pre-trained, quadratic-attention teacher model, which may limit its generalization.

**Audience:**

Yes

**Audience Explanation:**

- The paper proposes an effective and practical solution to enhance the efficiency of VLMs.

**Claims And Evidence:**

Yes

**Claims Explanation:**

The paper's central claims are generally supported by clear and convincing evidence.
-  The main contribution of the paper "LinMU achieves linear complexity while maintaining the performance of the teacher model" is supported by Table 1&5 (for comparison with teacher models and other VLMs)  and Figure 6&7 (for improved efficiency)

- The effectiveness of the proposed dual-branch M-MATE is supported by results in Table 4.

- The effectiveness of the proposed three-stage distillation is supported by results in Table 2.

**Requested Changes:**

-  The paper reports a total distillation time of ~50 hours on 8 A100 GPUs. While the distillation stage already involves a large amount training data, it would be better to compare the current setting with training LinMU with such data without teacher model. Also, it would be worthwhile to compare the training costs of distillation and standard training recipe of VLM.

---

> ### Author Response · Authors · 2026-03-02
>
> We thank the insightful comments from reviewer 7owK. Regarding the comments and requested changes, we respectfully respond as follows:
> 1. The cost of training a VLM from scratch. It is extremely expensive to train a powerful VLM from scratch and is not affordable for most researchers in academia. NVILA initializes its vision encoder from SigLIP and its LM decoder from Qwen2, but it still uses 208 H100 GPU days to complete training. Qwen2.5-VL-7B initializes its LM decoder by Qwen2.5 LLMs and still uses about 4.1T tokens to train its LM decoder, which requires at least thousands of H100 GPU days. For comparison, our fine-tuning cost is only 400 A100 hours, which approximately equals 5.6 H100 GPU days. Without the teacher model, 5.6 H100 GPU days are far from enough to complete even the very first stage of training a VLM. We added the analysis to a new subsection named “Training Cost Analysis”.
> 2. About the reliance on the powerful teacher model. On the one hand, our linear student can outperform the teacher model on some benchmarks, such as Video-MME, as shown in Table 1 of our paper. It is not capped by the teacher’s capabilities. On the other hand, training a powerful VLM from scratch is extremely expensive and not affordable for most academic researchers, as discussed in detail in the point above.

---

### Review · Reviewer_azyg · 2026-02-16

**Summary Of Contributions:**

The paper proposes a method to transform a pretrained multimodal model (image-text) into one with linear complexity with respect to input length. The approach replaces self-attention layers in the language model backbone with modules composed of two parallel branches: a bidirectional Mamba2 block (Flex-MA) and a fixed-window-size Swin attention (Local-Swin). To maintain performance, the authors employ a three-stage progressive distillation process from the original "teacher" model to the modified "student" model: first finetuning the Flex-MA branch, then both branches, and finally the whole model via LoRA. The method is evaluated on both image and video understanding benchmarks.

**Strengths:**

- The method significantly improves efficiency and token throughput for large VLMs (8B parameters), scaling better with input length than standard transformer architectures.
- It effectively preserves downstream performance compared to the teacher model used.

**Weaknesses:**

- There are significant concerns regarding the accuracy of the "linear complexity" claims, particularly regarding the vision encoder and existing literature.
- The evaluation lacks comparisons to other linear-complexity baselines, relying solely on comparing to the teacher model.

**Audience:**

Yes

**Audience Explanation:**

Efficient processing for Large Vision-Language Models (VLMs) is a highly relevant topic for the TMLR audience. The ability to improve token throughput and scale to longer inputs is critical for practical deployment. If the claims are clarified, the distillation strategy and architecture modifications (Mamba2 + Swin) would be of significant interest to researchers working on efficient architectures and to practitioners.

**Broader Impact Concerns:**

No specific ethical concerns were identified in the draft. The work focuses on architectural efficiency for existing open models.

**Claims And Evidence:**

No

**Claims Explanation:**

While the empirical results regarding performance preservation are convincing, the claims regarding **linear complexity** and **novelty** are not fully supported by the evidence provided:

1. **Complexity Claims:** The authors claim to achieve "pure linear complexity in input length." However, the method appears to leave the vision encoder untouched. Vision encoders also typically rely on attention blocks that scale quadratically with input length. Therefore, the _entire_ model effectively remains quadratic, even if the language model part is linear. Figure 6 supports this observation: while efficiency improves over the teacher, the correlation with input length does not appear strictly linear.
2. **Novelty Claims:** The paper claims to be "the first multimodal model that achieves pure linear complexity." This contradicts the paper's own related work section, which cites other VLMs that replace attention layers, like VL-Mamba (Qiao et al., 2024).
3. **Comparative Evidence:** The distilled models are compared exclusively to the original teacher model. To support claims of efficacy, the method should also be compared against other approaches that replace attention with linear-complexity operations.

**Requested Changes:**

**Major Changes:**

1. **Rectify Complexity Claims:** The authors must accurately qualify the "linear complexity" claims. If the vision encoder retains quadratic attention, the text must reflect that only the language backbone is linear, or the "pure linear" claim must be removed.
2. **Correct Novelty Statement:** Remove or modify the claim of being "the first" to achieve this complexity, specifically acknowledging the prior work cited in the related work section.
3. **Expand Baselines:** Include comparisons (performance and efficiency) against other methods that replace attention with linear-complexity operations, rather than comparing solely against the teacher model. This will help put in perspective the effectiveness of the proposed architecture.

**Minor Changes:**

4. **Unify Presentation:** The evaluations for the Qwen teacher setup and NVILA setup appear to use identical benchmarks (comparing Table 1 and Table 5). I recommend merging these or clarifying the distinction to improve readability. In Figure 7, please change the x-axis to represent the "number of tokens" directly rather than the "number of video frames," to allow for easier interpretation of efficiency scaling. Similarly, figure 7 could also be merged with figure 6.

---

> ### Author Response · Authors · 2026-03-02
>
> We thank the insightful comments from reviewer azyg. Regarding the mentioned weaknesses and requested changes, we respectfully respond as follows:
> 1. About the complexity of the vision encoder and our complexity claims. We admit that the Vision Encoder is left unchanged and possibly involves some full-attention layers, which have quadratic complexity. We made our claim more precise and added a “Limitation” section to discuss this. Based on the current results on the LM decoder, we speculate that we can also replace the global self-attention layers in the vision encoder with our proposed M-MATE blocks and distillation pipeline, and we leave it as part of future work. It is worth noting that: (1) The complexity of the vision encoder only affects TTFT, and the token throughput, which is usually more important for rapid interactions, is solely determined by the complexity of the LM decoder. (2) The latency of the vision encoder in lots of modern VLMs is already almost linear with respect to the number of patches, benefiting from efficient attention kernels like FlashAttention. Specifically, the vision encoder of Qwen2.5-VL has 31 layers with window attention and only one layer with full attention. We also measured the latency of the vision encoder in NVILA under different numbers of input video frames, as shown in Table 7. The latency is also almost linear with respect to the number of frames (i.e., the number of patches).
> An additional clarification regarding Fig. 6 is that the x-axis (i.e., the number of tokens) is increasing exponentially while the y-axis is linear. Thus, the TTFT curve of LinMU is not straight, even if the latency scales linearly with the number of tokens. The token throughput of LinMU remains unchanged when the input length increases, which supports our linear scaling claim.
>
> 2. About the novelty claim. When we make the claim of “the first”, we would like to emphasize that we are the first to achieve linear complexity without sacrificing model performance, while all the existing linear-complexity works incur performance degradation, especially when dealing with high-resolution images or long videos. This is supported by both the comparisons to teacher models and the comparisons to other linear-complexity or quadratic-linear-hybrid baselines.
>
> 3. About the comparison with other linear-complexity baselines. Lots of existing linear-complexity works didn’t train their model on video understanding tasks and are not large enough to perform fair comparisons (e.g., the largest size is a few hundred millions of parameters). To the best of our knowledge, we collected some linear-complexity and quadratic-linear-hybrid VLMs as the baselines and added them to Table 1. With similar or larger sizes, their performance lags behind NVILA and our LinMU student.
>
> 4. About unifying the presentation. For the Qwen teacher model, we conducted a subset of the full evaluation set to prove the generalization ability of our proposed framework and consider it as an ablation study. Thus, we didn’t merge Table 6 into Table 1. The temporal compression rate of Qwen is different from NVILA (2 vs. 8), so the mapping between the number of video frames and the input sequence length is different. Thus, we didn’t merge Fig. 7 into Fig. 6. We supplemented the mapping between the number of video frames and the input sequence length in the caption of Fig. 7.

---

### Decision · Action_Editor_Ly4H · 2026-04-08

**Recommendation:** Accept as is

**Additional Comments:**

All three reviewers recommend acceptance (one Accept, two Leaning Accept). The core contribution is clear: a practical framework for converting quadratic-attention VLMs into linear-complexity models while preserving performance, validated across multiple backbones and benchmarks.

The authors have been responsive to reviewer feedback, adding revisions such as linear-complexity baselines. These revisions substantively improved the paper.

All three reviewers agreed on audience interest, and the paper's practical efficiency gains make the findings relevant beyond the purely architectural research community.

**Audience:**

Yes

**Audience Explanation:**

The paper addresses a problem of broad and growing importance: the quadratic complexity bottleneck of self-attention in Vision-Language Models, which directly limits their deployment on edge devices and their ability to process high-resolution images and long-form videos. This is a core scalability concern for the multimodal ML community.

**Claims And Evidence:**

Yes

**Claims Explanation:**

The submission's claims are well-supported by accurate and clear evidence, particularly after the authors' revisions in response to reviewer feedback.